# Trends, geographic distribution, and disease burden of bipolar disorder in Ecuador (2011–2021): An analysis of hospital discharge data

Alberto Rodríguez-Lorenzana[1⊙], Marco Coral-Almeida[2,3⊙], Sarah J. Carrington[4⊙], Mabel Torres-Tapia[5‡], Diana Álvarez-Mejía[5‡], Milena Santana[2‡], Guido Mascialino[5*]

**1** Health Sciences Department, Universidad Pública de Navarra, Navarra, Spain, **2** Veterinary Medicine Department, Universidad de las Américas, Quito, Ecuador, **3** Biochemoinformatics Group, General Direction of Research and Liaison, Universidad de las Américas, Quito, Ecuador, **4** Pontificia Universidad Católica del Ecuador, Quito, Ecuador, **5** CEC Research Group, Universidad de las Américas, Quito, Ecuador.

⊙ These authors contributed equally to this work.
‡ These authors also contributed equally to this work.
* guido.mascialino@udla.edu.ec

## Abstract

This retrospective observational study aims to evaluate the incidence, disease burden, and geographic distribution of bipolar disorder based on hospital records in Ecuador over an eleven-year span. Hospital discharge data, publicly available from 2011 to 2021, were analyzed to assess incidence, DALYs, and the spatial distribution of hospitalized cases during this period. Between 2010 and 2021, a total of 6,821 hospitalized cases of bipolar disorder were documented in Ecuador, comprising 2,423 males and 4,398 females. The incidence rate peaked in 2019, with the lowest rate reported in 2020. There was no linear association between time and incidence rates or number of cases, but a significant increase was observed from 2017 to 2019 (p < 0.0001). The incidence rate was significantly higher in females compared to males (p < 0.0001). The average annual incidence was 3.47 cases per 100,000 person-years. The mean age at diagnosis was 40.76 years, with females being diagnosed at a younger age than males (p = 0.01548). Bipolar disorder-related deaths totaled 27 (12 males, 15 females). The burden of disease, expressed in DALYs, ranged from 66.769 to 126.98 per 100,000 population, with the hospitals from the private sector contributing most to the average DALYs. YLDs represented over 99% of the total burden. This study highlights the significant gender differences and temporal trends in bipolar disorder incidence in Ecuador, emphasizing the need for targeted public health strategies.

## Introduction

Bipolar disorder is a complex mood disorder with a multifactorial origin and uncertain etiology, impacting 1–2% of the global population [1,2]. It is a major contributor to

**Data availability statement:** The data that support the findings of this study are publicly available and can be accessed at the following links: https://www.ecuadorencifras.gob.ec/camas-y-egresos-hospitalarios/ (Descriptive data on hospital discharges and bed occupancy); https://www.ecuadorencifras.gob.ec/proyecciones-poblacionales/ (Population projections, used for incidence calculations). These datasets are freely accessible and were used in accordance with the terms and conditions specified by the respective sources. No proprietary or restricted data were used in this study.

**Funding:** The author(s) received no specific funding for this work.

**Competing interests:** The authors have declared that no competing interests exist.

global disability, significantly impairing the social functioning of those affected [3]. Furthermore, it is linked to a shorter life expectancy compared to the general population [4], collectively imposing a substantial societal burden.

While bipolar disorder often emerges in early adulthood, diagnosis is frequently delayed by several years. Some individuals report delays of up to a decade from symptom onset to diagnosis [5], attributed to the alternation between emotional extremes [6] and the variability in life trajectories among affected individuals [7]. These factors complicate accurate and timely diagnosis, and lack of treatment can worsen the disease's course and prognosis.

Bipolar disorder is associated with adverse health outcomes, including elevated mortality rates. Premature death in individuals with this condition is frequently linked to comorbid medical conditions such as cardiovascular disease, metabolic disorders, obesity, and diabetes [3]. Suicide also significantly contributes to mortality rates, accounting for 15% of deaths in this population [8].

Gender has also been shown to have an important influence on symptom presentation, comorbidity, and response to bipolar disorder treatment. Women tend to experience more depressive episodes and comorbidities such as anxiety and eating disorders, while men experience more manic episodes and substance abuse [9,10]. These differences also affect their response to treatment, with women being more susceptible to the side effects of mood stabilizers and men showing higher rates of completed suicidal behavior [11,12]. In Ecuador, however, little research has been undertaken to analyze these disparities, making it difficult to develop public health interventions that address gender-specific needs [13].

Given the severity and chronic nature of bipolar disorder, its economic impact is substantial. In Europe, the estimated total cost of the condition is approximately 21.49 billion euros, whereas in the United States, it reaches 151 billion dollars. Insurance studies reveal that bipolarity is one of the costliest disorders compared to other mental and medical conditions, second only to diabetes combined with coronary heart disease. Depressive symptoms lead to high indirect costs due to loss of productivity, while manic symptoms significantly influence direct costs due to hospitalizations [3].

Epidemiological data and information on the disease burden of bipolar disorder in Latin America remain scarce. In Brazil, the years lived with disability (YLDs) due to bipolar disorder are nearly 249 per 100,000 inhabitants, representing the highest rate in Latin America and the Caribbean. Paraguay and Argentina follow with 245 and 234 years, respectively [14]. The lifetime prevalence of bipolar disorder in Latin America is estimated at 2.3% [15]. However, the genetic and social risk factors influencing the disorder's clinical presentation and associated comorbidities remain insufficiently studied.

Bipolar disorder imposes a significant economic burden in Latin America, stemming from both direct and indirect costs. Direct costs encompass hospitalizations, medications, and outpatient care, often aggravated by the region's limited mental health infrastructure. In Brazil, for example, nearly $9 billion is spent annually on managing bipolar disorder, a figure compounded by the fact that nearly half of patients remain symptomatic despite treatment, resulting in recurrent episodes and increased hospitalizations [7,16].

Indirect costs, including absenteeism, loss of productivity, and societal burden, are equally significant. Bipolar disorder is among the leading causes of disability-adjusted life years (DALYs) in the region, with tropical Latin America reporting the highest incidence and DALY rates globally [17,18]. Comorbidities, such as anxiety and substance use disorders, further increase healthcare utilization and complicate disease management, inflating the overall economic strain [19,20].

The treatment gap for bipolar disorder exceeds 80% in countries like Colombia, delaying diagnosis and exacerbating disease severity, which leads to more costly interventions, such as hospitalizations [21,22]. In low- and middle-income countries like Ecuador, these pressures are amplified by under-resourced mental health systems characterized by inequitable access, outdated infrastructure, and a shortage of trained professionals [23]. Furthermore, the indirect costs of bipolar disorder, particularly in low-income populations, create substantial societal and financial disruptions, as these groups lack the financial resilience to absorb such impacts [24]. Addressing these challenges requires urgent public health interventions, such as expanding access to affordable mental health services, implementing early diagnostic and treatment strategies, and closing the treatment gap to reduce the broader economic and social consequences of bipolar disorder.

Ecuador lacks specific epidemiological studies on bipolar disorder, making it challenging to accurately determine the disease burden. Global statistics suggest that it is one of the costliest and most burdensome diseases. However, the impact of it on Ecuador's public health services remains unknown, complicating the establishment of effective treatment plans at different levels of care. Studies like this one can offer essential evidence to support the distribution of resources and the implementation of public health initiatives focused on the prevention and treatment of bipolar disorder [23].

This research seeks to fill the existing knowledge gap by examining data from individuals diagnosed with bipolar disorder and hospitalized in Ecuador. It focuses on exploring the characteristics of the disorder, with particular attention to gender disparities and its impact on the population at the national level. The ultimate goal is to enhance decision-making processes related to advancing strategies for awareness, prevention, and management of this mental health condition in Ecuador.

## Methods

A dataset publicly provided by the National Institute of Statistics and Censuses (INEC), a state agency tasked with compiling nationwide statistical information, including hospital data from across the country, was analyzed. This dataset encompasses anonymized information related to hospital discharges, encompassing both public and private healthcare facilities. This information is in the public domain and therefore no ethics committee clearance was required. The INEC is responsible for data collection across multiple domains that inform public policy. It ensures data quality through a governance framework that includes automated systems for secure and consistent data handling, rigorous validation protocols, and adherence to international standards. Processes such as consistency checks, de-duplication, and collaboration with health institutions resolve discrepancies, ensuring reliable and comprehensive data for public health policy. The anonimity of the database precludes consideration of repeated hospitalizations, which are a limitation of the data.

The dataset, spanning the years 2010–2021, included demographic variables such as age, sex, self-reported ethnic or racial identity, and discharge diagnoses. The discharge diagnoses were classified in the database following the guidelines of the 10th Revision of the International Classification of Diseases (ICD-10). Diagnoses identified as Bipolar Affective Disorder corresponded broadly to the ICD-10 classification codes F31. which include F31.0 Bipolar affective disorder, current episode hypomanic; F31.1 Bipolar affective disorder, current episode manic without psychotic symptoms; F31.2 Bipolar affective disorder, current episode manic with psychotic symptoms; F31.3 Bipolar affective disorder, current episode mild or moderate depression; F31.4 Bipolar affective disorder, current episode severe depression without psychotic symptoms; F31.5 Bipolar affective disorder, current episode severe depression with psychotic symptoms; F31.6 Bipolar affective disorder, current episode mixed; F31.7 Bipolar affective disorder, currently in remission; F31.8 Other bipolar affective disorders; and F31.9 Bipolar affective disorder, unspecified [25]. As of 2021, Ecuador's population was estimated at 17,751,277, compared to 15,012,228 recorded during the most recent nationwide census in 2010 [26].

## Sources of information

In this investigation, information spanning the years 2010–2021, sourced from national-level records documenting hospital deaths and admissions resulting in discharges and related to specific illnesses, was employed. These registries are compiled by the General Direction of Civil Registry and the Ministry of Health in Ecuador and subsequently aggregated and disseminated by Instituto Nacional de Estadística y Censos (INEC). The dataset utilized in this research was extracted from these consolidated databases, encompassing data from both public and private healthcare service providers [26].

The database provided details of the sources of hospital record information categorized by the type of institution. The public sources listed include Seguro Social (Social Security), the Instituto Ecuatoriano de Seguridad Social (Ecuadorian Social Security Institute), the Junta de Beneficencia de Guayaquil (Guayaquil Charitable Board), the Ministerio de Defensa Nacional (Ministry of National Defense), the Ministerio de Educación (Ministry of Education), the Ministerio de Gobierno y Policía (Ministry of Government and Police), the Ministerio de Justicia y de Gobierno y Policía (Ministry of Justice and Government and Police), the Ministerio de Justicia, Derechos Humanos y Cultos (Ministry of Justice, Human Rights, and Worship), the Ministerio de Salud Pública (Ministry of Public Health), municipalities, other public institutions, Seguro Campesino (Rural Social Security), and universities and polytechnic schools. The private sources are grouped into two categories: private institutions for profit and private institutions not for profit [27].

The international ICD codes F31.0 – F31.9 were applied for the identification of fatalities and hospital admissions attributed to Bipolar Affective Disorder. Prior to employing the DALY calculator in R v4.1.2 [28], the data underwent processing in Microsoft Excel.

## Estimation of the burden of disease

To assess the impact of Bipolar Affective Disorder on the overall disease burden in Ecuador during the 2010–2021 period, Disability-Adjusted Life Years (DALYs) were estimated. This metric combines the disease burden derived from both the duration of disability experienced by individuals (YLDs) and the years of life lost due to early mortality (YLLs). The approach followed the methodology outlined by Murray et al. [29,30] in the Global Burden of Disease (GBD) studies. Calculations were performed using the "DALY" package for R [28].

Firstly, to estimate Years of Life Lost (YLLs), the study calculated the product of the number of deaths attributed to Bipolar Affective Disorder during the study period and the remaining life expectancy at the age of death. Residual life expectancy was determined using the R software, based on GBD 2010 data, with a standardized life expectancy of 86.02 years applied to both males and females. For the DALY estimations, three distinct methodologies were utilized: the first excluded both a time discount rate and age weighting for the years lost; the second incorporated a 3% annual time discount rate, emphasizing years nearer to the present while omitting age weighting; and the third combined the 3% annual discount rate with age weighting. This approach adheres to the framework described by Murray et al. [29] and Egunsola et al. [31].

Secondly, to complete the calculation of the total burden of disease, we need to add to the YLL the Years Lived with Disability (YLD), which is the more substantial source of disease burden with Bipolar Affective Disorder. Effectively the YLD reflects the size of the gap between the expected quality of life lived without the illness and that lived with it. It is important to highlight that the hospital admission data recorded by healthcare services reflect only the segment of the symptomatic bipolar population with sufficient access to medical care. Consequently, these data are not appropriate for estimating the total Years Lived with Disability (YLDs) at the national level but are instead utilized to estimate the YLDs observed within the healthcare system.

To calculate YLD estimates, it is essential to possess Disability Weights (DW) estimates specific to the diagnosed population with Bipolar Affective Disorder. The DW offers an understanding of the anticipated effect of the illness on quality of life, based on available data regarding the typical severity of disability experienced by individuals diagnosed with the condition. Given the absence of detailed information on the severity of Bipolar Affective Disorder experienced by individual

patients in the sampled hospital records, utilizing the average disability weight is regarded as the most unbiased assumption. Accordingly, we inferred the distribution of the severity of experience within the population and took this distribution to calculate the weighted average DW for sufferers of Bipolar Affective Disorder.

According to Kim et al. [32], the DW for mild, moderate and severe Bipolar Disorder are 0.248, 0.453 and 0.658, respectively. The same authors also estimated a distribution of the severity of illness amongst the diagnosed population of 9%, 61% and 30%, for mild, moderate and severe Bipolar Affective Disorder, respectively. Using the available data, we computed a weighted average disability weight of 0.496, which was then applied consistently across all cases in our database.

The age-specific data reveals that the incidence rate of schizophrenia diagnosis varies across different age groups, ranging from an average of 0 per 100,000 person-years for infants of both sexes to 4.8 per 100,000 individuals (males) in the 45–59 year age group, and 8.6 per 100,000 individuals (females) for adults in the same age group. Table 1 presents the details of the parameters used to estimate the disease burden, along with their corresponding probability distributions, which were integrated into the DALY package for R.

## Statistical Analyses

The study gathered descriptive statistics for all variables. The statistical significance was evaluated with Pearson´s Chi-squared regressions, Poisson, linear regressions, and significance was accepted with p-values of less than 0.05 to analyze the difference between year and gender. Statistical significance was determined using 95% Poisson confidence intervals. The analyses and figures were carried out in R version 4.1.3 (2021) and Microsoft Excel.

## Spatial Analyses and Statistical Methods

Spatial analyses were used for the identification of spatial clusters with significant incidence rates for bipolar disorder in each canton. The cases were distributed by canton and ICD-10 identification code to identify those cantons with a statistically significant spatial cluster from 2010 to 2021.

The study´s spatial analyses found areas with statistically significant higher incidence rates by using SATSCAN v10.1.2, the most recent version as of May of 2023 [33]. The applied methodology was previously described by Ron-Garrido et al. [34] and Kulldorff [35]; with minor modifications to conduct spatial-only analysis. The population, number of

**Table 1. Cases of bipolarity from 2010 to 2021.**

| Year | Number of cases | Persons-time at risk | Incidence rate in 100,000 person-years | Poisson confidence intervals at 95% |
|---|---|---|---|---|
| 2010 | 513 | 15012228 | 3.42 | [3.13;3.73] |
| 2011 | 493 | 15266431 | 3.23 | [2.95;3.53] |
| 2012 | 495 | 15520973 | 3.19 | [2.91;3.48] |
| 2013 | 456 | 15774749 | 2.89 | [2.63;3.17] |
| 2014 | 560 | 16027466 | 3.49 | [3.21;3.80] |
| 2015 | 566 | 16278844 | 3.48 | [3.20;3.78] |
| 2016 | 570 | 16528730 | 3.45 | [3.17;3.74] |
| 2017 | 687 | 16776977 | 4.09 | [3.79;4.41] |
| 2018 | 708 | 17023408 | 4.16 | [3.86;4.48] |
| 2019 | 727 | 17267986 | 4.21 | [3.91;4.53] |
| 2020 | 490 | 17510643 | 2.80 | [2.56;3.06] |
| 2021 | 556 | 17751277 | 3.13 | [2.88;3.40] |
| Total | 6821 | | | |
| Yearly Mean Incidence | | | 3.47 | [3.39;3.55] |

cases and the geographic coordinates of each canton were used in the spatial-only analysis. To compare the number of cases in each area a Poisson distribution was used.

The space clustering was assessed by contrasting the likelihood ratio which determined the significance of the identified space clusters. P-values were calculated by applying 999 Monte Carlo simulations. Spatial clusters with a p-values below 0.05 were determined as significant. The Gini coefficient was applied to narrow down these clusters as described by Han et al. [36]. Further spatial analyses were carried out in QGIS version 3.8 Zanzibar software. The maps were designed and created by the authors. The shape files used in this study for province and cantonal borders were obtained from geoBoundaries: A global database of political administrative boundaries (www.geoboundaries.org) [37]. Continental and country borders were Made using shapefiles obtained from Natural Earth portal http://www.naturalearthdata.com/

Population estimates were adjusted according to year and gender to modify the incidence rates during the study period. The incidence rates are given in terms of both absolute new case count and relative rates per 100,000 people. The epitools package in R was used to apply 95% Poisson confidence intervals. Linear regression models were used to establish further associations.

## Results

From 2010 to 2021, 6821 bipolar disease cases were recorded in Ecuador, where 2423 were male patients and 4398 were female. The highest incidence rate was reported in 2019 while the lowest was recorded in 2020 (Table 1). There was no linear association between time and incidence rates nor with number of cases (p-value = 0.492 and p-value 0.0884), however, for the years 2017, 2018 and 2019, there was a significant increase in the number of admitted cases when compared with other years (p-value<0.0001). There was a significant difference between males and females in the incidence of bipolar disorder (p-value<0.0001).

### Characteristics by sex

**Females.**  During the study period, 4398 female patients were diagnosed with bipolar disorder, where the year with the highest incidence rate was recorded in 2019, and the lowest one was registered in 2013. There was a significant positive association between time and incidence rates for females (p-value 0.047) which suggests an increase in bipolar disorder cases from 2010 to 2021, the years with the highest incidence and number of cases were 2017, 2018 and 2019 (p-value<0.01) (Fig 1).

**Males.**  From 2010 to 2021, a total of 2423 cases were diagnosed in male patients. The highest incidence rate was reported in 2017, meanwhile, the lowest was recorded in 2020. No associations were identified between time and incidence rates nor between time and the number of admitted cases (p-value >0.6 and p-value> 0.4 respectively). However, 2020 was the year with the statistically lowest incidence of bipolar disorder (p-value<0.0013) (Fig 1).

### Characteristics by age and sex

Additional analyses were made to compare the age when patients were admitted with bipolar disorder (Fig 2). The mean age was 40.76 with Poisson confidence intervals at 95% [40.39; 41.13]. The mean age of diagnosis for female patients with bipolar disorder was 40.42 with Poisson confidence intervals at 95% [39.96; 40.87]. For male patients, the mean age of diagnosis for bipolar disorder was 41.38 with Poisson confidence intervals at 95% [40,75;42.00]. There was a significant difference between the mean ages of admission for males and females (p-value 0.01548). Female patients were diagnosed at a younger age with bipolar disorder.

### Burden of disease expressed in DALYs

During the study period, a total of 27 patients were recorded by authorities to have died from the Bipolar Affective Disorder. A total of 12 deaths corresponded to male patients and 15 to female patients.

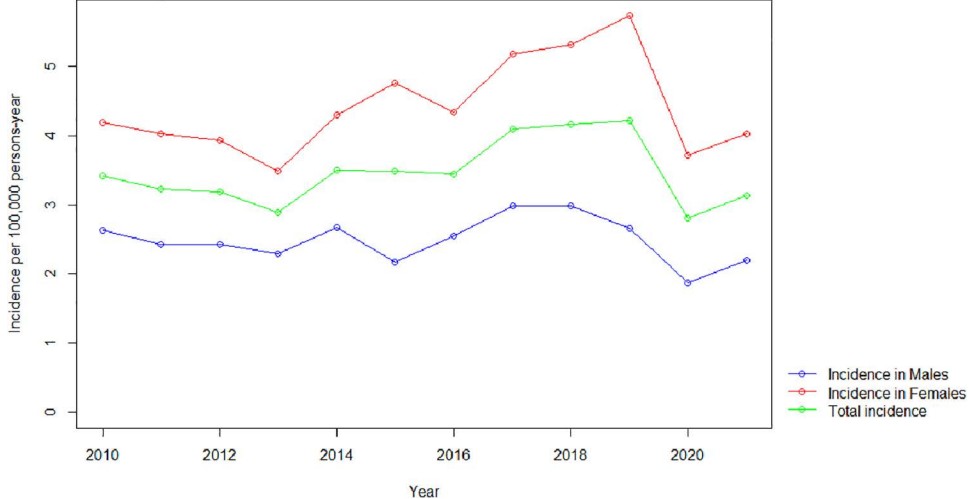

**Fig 1. Time trend of bipolar disorder hospitalized yearly incidences male and female cases (2010-2021).** The graph shows the annual incidence of hospitalizations for bipolar disorder per 100,000 person-years, with separate trends for males (blue line), females (red line), and the total population (green line). Female incidence rates are consistently higher than those of males across the entire period. The total incidence follows a similar trend to females, reflecting their predominant contribution. A notable peak is observed in 2019, followed by a decline in subsequent years. These trends provide insights into sex-based differences in hospitalization rates for bipolar disorder over time.

The estimated burden of disease of Bipolar Affective Disorder in Ecuador (2010–2021) varied from 66.769 to 126.98 per 100,000 population on average depending on the scenario used for estimation (discount rates and age weighting scenarios, as above escribed). Hospital treatment in the private sector contributed with the highest average proportion of DALYs per sector. Hospital care in the public sector was the group with the lowest DALY average contribution. The results are detailed in Table 2.

YLDs represented over 99% of all contributions to the burden of disease in DALYs for all sectors.

### Spatial distribution of bipolar disorder in Ecuador

Fig 3 shows five spatial clusters with significant augmented risk of hospitalization due to bipolar disorder. While Fig 4 represents the spatial distribution of the incidence of bipolar disorder hospitalized cases.

Five significant spatial clusters for augmented risk of hospitalization due to bipolar disorder were identified. Four out of five clusters were identified in provinces located in the highlands of Ecuador, and one in one coastal province. The cluster with the highest relative risk was identified in the province of Pichincha where the capital city is located.

### Discussion

This is the first study to perform a sociodemographic characterization of the incidence and disease burden of bipolar affective disorder in Ecuador using national hospital records. The incidence of hospitalized cases for all groups between 2010–2021 was 3.47 per 100,000 persons per year. The lowest incidence reported in this period was in 2020 with 2.80 per 100,000 persons per year, likely due to the COVID-19 pandemic. The pandemic led to significant declines in accessibility to medical care because of quarantine measures, hospital restrictions, and competition for limited resources [38].

Mental health was particularly affected in middle and low-income countries like Ecuador, where inadequate funding has resulted in a disproportionate distribution of professionals to the sector, as well as insufficient training, and outdated infrastructure [39]. Other studies have also reported a decrease in bipolar disorder diagnoses in hospitals since 2019 [,40,41],

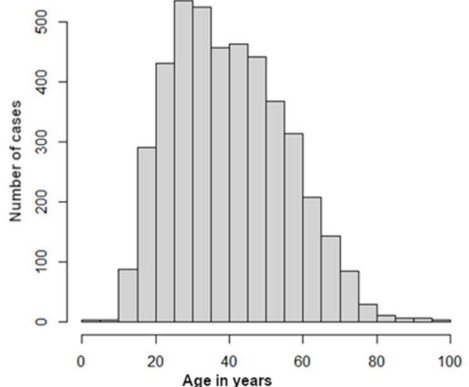
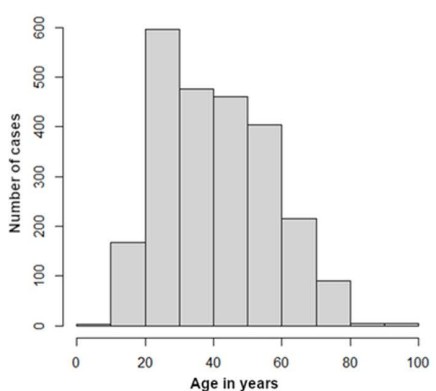

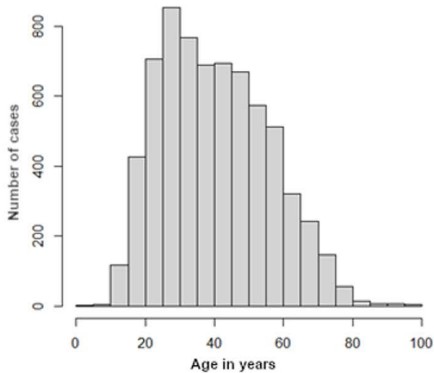

**Fig 2. Distribution of Bipolar disorder hospital admission by age (total, female and male).** He histograms show the number of hospitalized cases across age groups for the total population (bottom), females (top-left), and males (top-right). All distributions demonstrate a peak in hospitalization rates among individuals aged 20–40 years, with a gradual decline in older age groups. The female distribution shows a slightly higher number of cases compared to males, particularly in younger and middle-aged groups, reflecting sex-based differences in hospital admissions for bipolar disorder.

**Table 2. Burden of disease of Bipolar Affective Disorder expressed in DALYs.**

|  | Total (Public and Private sector) | Public Sector | Private Sector |
|---|---|---|---|
| DALY/POP/100000 |  |  |  |
| No age weighting and no discount rate | 126.980 | 58.003 | 69.208 |
| Age weighting and no discount rate | 113.890 | 52.028 | 62.002 |
| Age weighting and 3% discount rate | 66.769 | 30.508 | 36.367 |
| Contributions of YLD and YLL for DALYs |  |  |  |
| YLD/DALY | 99.42% | 99.50% | 99.42% |
| YLL/DALY | 0.58% | 0.50% | 0.58% |

consistent with our findings, possibly reflecting a lack of access to mental health care rather than an actual decrease in incidence [42].

These findings highlight the importance of developing public policies focused on improving mental health systems in Ecuador. This is especially relevant in the current context, where events such as the COVID-19 pandemic have demonstrated the urgency of including mental health as a priority in responses to health emergencies.

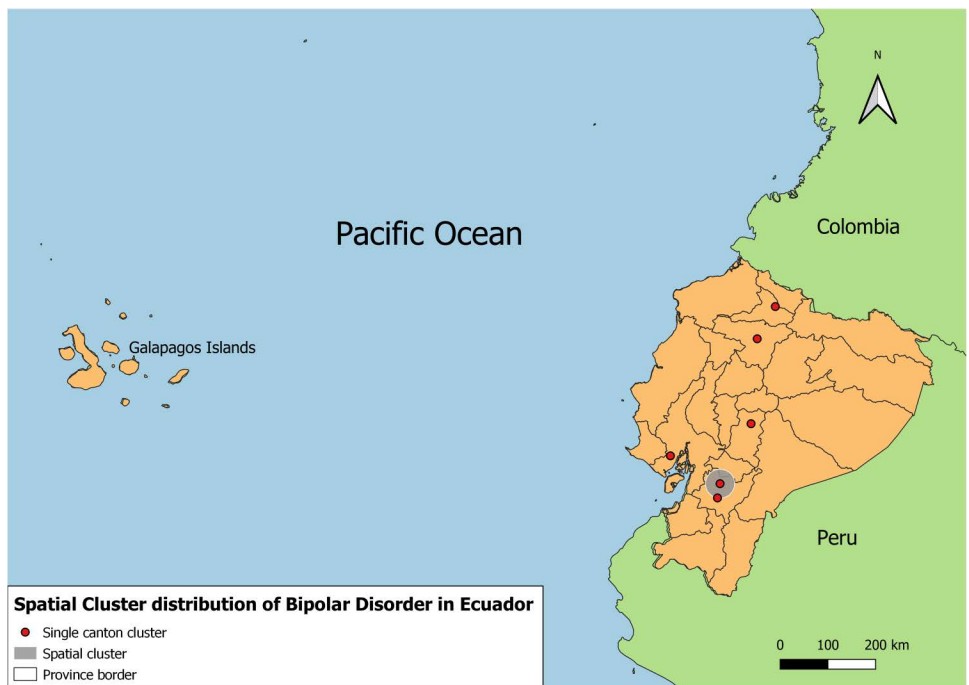

**Fig 3. Spatial clusters of bipolar disorder in Ecuador.** This figure illustrates spatial clusters of bipolar disorder incidences across Ecuador, identifying regions with significantly higher or lower incidence rates. Clusters were determined using SaTScan, based on data from 2010-2021. Single cluster cantons with statistically significant clusters are marked in red dots for high-incidence clusters, grey circles describe high incidence larger areas including more than one single canton cluster. The shape files used in this figure for province and cantonal borders were obtained from geoBoundaries: A global database of political administrative boundaries [37]. Continental and country borders were Made using shapefiles obtained from Natural Earth portal http://www.naturalearthdata.com/.

Our study found a significant positive association between time and incidence only in women, indicating an increase in cases over the studied period. This aligns with global data showing an increase in the incidence and prevalence of bipolar disorder over the years [43–45]. This trend can be attributed to demographic changes, such as increased life expectancy and an older average age [46], as well as greater diagnostic awareness and attention towards bipolarity since 2000 [47].

The Global Burden of Diseases (GBD) study determined that the age-standardized prevalence of bipolar disorder per 100,000 people for Latin America and the Caribbean was 963.7, in Andean Latin America 910.5, in Central Latin America 854, and in Tropical Latin America 1111.1 [48]. These figures suggest that Ecuador has a low registration of cases, possibly influenced by the treatment gap in the region. For instance, in Colombia, the treatment gap for affective disorders is 84.8% [21], indicating that many individuals do not receive necessary care for diagnosis and treatment.

Reducing this treatment gap should be a priority for the Ecuadorian health system. Measures that have been shown to be effective in addressing this problem include: 1) the development of subsidy programmes or economic incentives to facilitate access to mental health services for low-income groups, 2) the promotion of initiatives to extend health coverage to remote areas with limited access to specialized services, and 3) the strengthening of education and training of specialized health personnel.

Sociodemographic variables, such as sex, is crucial for understanding the diagnosis and course of this disorder. Although some studies suggest equal prevalence in men and women [49,50], our study found significant disparities. There were more cases in females (4,398) than in males (2,423), with an average incidence rate of 4.43 vs. 2.48 per 100,000 persons per year. This may be due to better diagnosis for women related to their higher risks, especially among pregnant and postpartum women, for bipolar disorder type II, hypomania, rapid cycling, and mixed episodes [49,51].

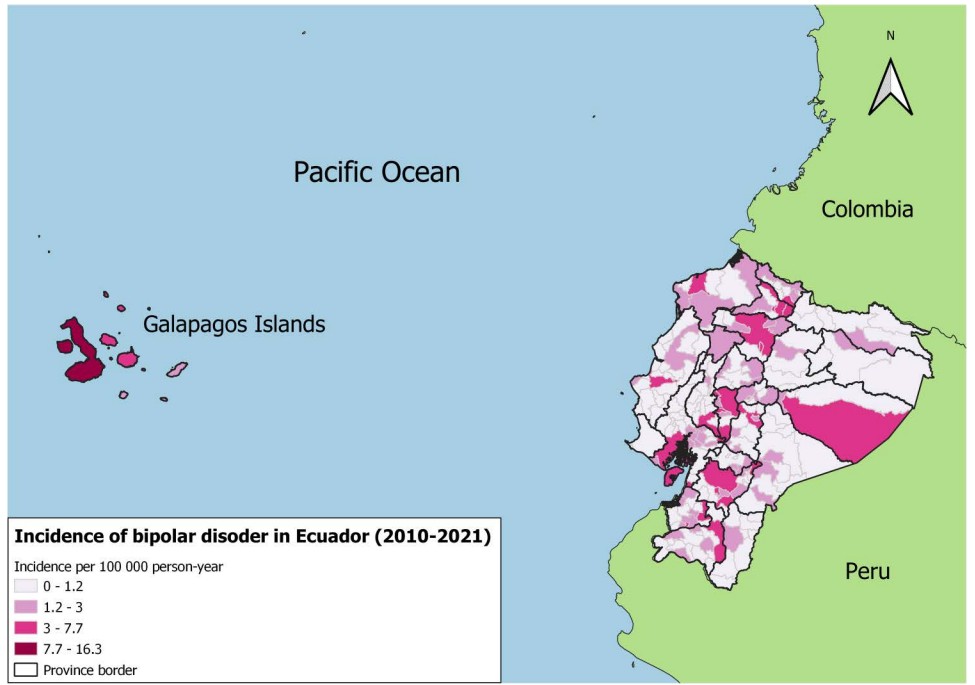

**Fig 4. Incidences of bipolar disorder in Ecuador (2010–2021).** The figure depicts the distribution of bipolar disorder incidences across different regions of Ecuador for the 2010–2021 period. The shape files used in this figure for province and cantonal borders were obtained from geoBoundaries: A global database of political administrative boundaries [37]. Continental and country borders were Made using shapefiles obtained from Natural Earth portal http://www.naturalearthdata.com/.

Men typically have an earlier onset of first episode of mania than women, who have higher incidence rates of bipolar disorder type I across adulthood, though gender differences are not statistically significant [52]. In contrast, our study shows that the mean age of this diagnosis for men was 41.38 years and for women 40.42 years. The highest peak of diagnosis was between 25 and 30 years, with higher incidence in women. The relatively earlier diagnosis and higher recorded incidence rates for women may suggest that professionals are better at identifying the signs of the illness in women, or women are better at seeking help for their illness, or both. Incidence by sex in other age groups did not differ significantly.

There is a significant gap between the age of onset (usually late adolescence) and age of diagnosis. Delays in diagnosis range from 5 years in bipolar disorder type I to 11 years in type II in Canada, and on average 8.74 years in Australia [53]. In Chile and Argentina, the delay is around 8 years, while in Ecuador, it extends to 15–20 years. The difficulty in diagnosing BD, especially in the initial stage, contributes to this delay. Misdiagnosis as depression and the complexity of assessing manic or hypomanic symptoms during depressive periods further complicates timely diagnosis [54].

This delayed diagnosis leads to significant costs through prolonged suffering, and worse prognosis, contributing to bipolarity having the highest suicide rate among psychiatric conditions [24,12]. Patients often endure an average of 10 years with symptoms without a correct diagnosis, exacerbating the disease burden.

Reducing delays in diagnosis is essential to reduce the negative impact of bipolar disorder on patients' quality of life. The implementation of screening systems in primary care or the development of educational campaigns aimed at the general public could contribute to the recognition of early signs of the illness and reduce this gap, especially in adolescents and young adults.

The results also show that the private sector handles more of the burden from bipolar disorder (69.208) than the public sector (58.003). This may be because, as several studies have reported, patients with bipolar disorder tend to have a higher socioeconomic status [55,56]. People with higher resources and status tend to seek professional help and access better treatment [56] and in the context of the Ecuadorian health system, this usually is private. On the other hand, no differentiation in burden by sex was found in global studies [57].

Observing the geographical distribution of the incidence of bipolarity cases, the provinces with more cases are Pichincha, Pastaza, Bolivar, Guayas, Azuay and Galapagos. While the provinces with the most risk for hospitalization cases are Imbabura, Pichincha, Chimborazo, Azuay and Guayas. We can say that most cases and risk factors occur in the highlands and there is also a higher incidence in the provinces where the largest and most populated cities such as Quito, Guayaquil and Cuenca are located.

Regarding Disability-Adjusted Life Years (DALYs), the data indicates that from 2010 to 2021, there was a disease burden of 126.980 per 100,000 inhabitants in Ecuador. Global numbers suggest that this disorder resulted in 105.4 DALYs per 100 000 persons (95% UI 64–3–162-0) in 2019, a number in line with our results. Such numbers place bipolar disorder as 67th in the ranking of global causes of DALYs and as 28th for causes of YLDs [18]. As for Ecuador, it was found that YLDs explain 99% of all contributions to the burden of disease in DALYs for all sectors.

The findings from this study, which estimate a mean annual incidence of 3.47 per 100,000 person-years and a burden of disease ranging from 66.77 to 126.98 DALYs per 100,000 population in Ecuador, align with trends observed in other Latin American countries, albeit with notable variations. For instance, in Brazil, bipolar disorder accounts for approximately 249 years lived with disability (YLDs) per 100,000 inhabitants, the highest in Latin America, followed by Paraguay (245 YLDs) and Argentina (234 YLDs) [15].

The lifetime prevalence of bipolar disorder in Latin America is estimated at 2.3%, slightly higher than global figures [2]. However, epidemiological studies in countries such as Colombia and Chile reveal significant gaps in diagnosis and treatment. Colombia, for example, has a treatment gap of 84.8% for affective disorders, highlighting the regional challenges in accessing care [21].

The spatial clustering observed in Ecuador's highlands and major urban centers like Pichincha mirrors findings in Argentina and Brazil, where urbanization and socioeconomic factors contribute to the concentration of cases (GBD 2019 Mental Disorders Collaborators, 2022) [48]. These regional comparisons underscore the shared challenges across Latin America in addressing mental health disorders, including disparities in healthcare access, diagnostic delays, and the societal stigma surrounding mental illness.

## Limitations

This study provides valuable insights into the epidemiology and burden of bipolar disorder (BD) in Ecuador; however, it is constrained by several limitations that could impact the comprehensiveness of its findings. Firstly, the reliance on hospital discharge records means the analysis captures primarily severe cases requiring hospitalization, excluding undiagnosed or outpatient cases. This approach likely underestimates the true prevalence and burden of BD, particularly in rural and underserved areas where access to hospital care is limited. Additionally, the data pertain only to diagnosed and hospitalized cases, and the lack of comprehensive mental health services and social support systems in Ecuador further skews the dataset, as many individuals with BD may remain undiagnosed or untreated.

Furthermore, variability in diagnostic practices across institutions, coupled with potential inaccuracies in ICD-10 coding, may lead to inconsistencies in identifying and categorizing BD cases, particularly their subtypes. The management and quality of diagnostic registries also vary between public and private healthcare systems, which could contribute to coding inaccuracies and gaps in data. Moreover, data on hospital discharges may include repeat hospitalizations of the same individuals across different institutions, potentially leading to overrepresentation of certain cases. Conversely, the underrepresentation of outpatient and mild cases results in an incomplete picture of BD's true epidemiological footprint.

Cultural and socioeconomic factors likely influence mental health-seeking behaviors and access to treatment, further complicating efforts to gauge the full burden of BD. These factors disproportionately affect marginalized populations, such as those in rural areas or with limited financial means, resulting in potential underrepresentation in the dataset.

To address these limitations, future research should adopt a multifaceted approach. Community-based studies and household surveys can complement hospital data by capturing cases that remain undiagnosed or are managed outside of formal healthcare settings. Integrating data from public and private healthcare providers into a unified electronic health system could reduce duplication, improve case tracking, and provide a more comprehensive understanding of BD prevalence. Standardizing diagnostic criteria and training healthcare personnel in mental health diagnostics would enhance the accuracy and consistency of data collection. Furthermore, qualitative research exploring cultural and socioeconomic barriers to diagnosis and care would help illuminate gaps in understanding BD's impact across different populations.

Finally, longitudinal studies that follow patients from symptom onset to diagnosis would be invaluable in addressing the diagnostic delay frequently observed in BD cases. Advanced methodologies, such as geospatial analysis and machine learning, could uncover trends and risk factors, guiding targeted public health interventions. Such efforts would contribute to reducing the burden of bipolar disorder in Ecuador and similar resource-constrained settings.

## Conclusion

This study is the first to characterize the incidence and burden of bipolar disorder in Ecuador over an 11-year period. The findings indicate a low incidence of bipolar disorder, likely due to the treatment gap common in low-income countries. Notably, women are more frequently diagnosed with BD, though men tend to receive earlier diagnoses. The burden of disease in Ecuador aligns with global figures, underscoring the significant impact of BD. These results highlight the urgent need for public policies aimed at early diagnosis and comprehensive care to mitigate the personal and social impacts of bipolar disorder.

## Supporting information

**S1 Table. Parameters for DALY calculation.**
(DOCX)

**S2 Table. Female cases of bipolar disorder and incidence by year.**
(DOCX)

**S3 Table. Cases and incidence of bipolar disorder in male Ecuadorians by year.**
(DOCX)

**S4 Table. Spatial clusters of BD in Ecuador.**
(DOCX)

## Acknowledgments

We would like to thank Natalia Barriga for her help with curation of the data.

## Author contributions

**Conceptualization:** Alberto Rodríguez-Lorenzana, GUIDO MASCIALINO.

**Data curation:** Marco Coral-Almeida, Sarah J. Carrington.

**Formal analysis:** Marco Coral-Almeida, Sarah J. Carrington.

**Methodology:** Marco Coral-Almeida, Sarah J. Carrington, Milena Santana.

**Supervision:** Alberto Rodríguez-Lorenzana, GUIDO MASCIALINO.

**Writing – original draft:** Marco Coral-Almeida, Sarah J. Carrington, Mabel Torres-Tapia, Diana Álvarez-Mejía, Milena Santana.

**Writing – review & editing:** Alberto Rodríguez-Lorenzana, Mabel Torres-Tapia, Diana Álvarez-Mejía, GUIDO MASCIALINO.

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
