## [Decision Letter · Decision Letter 0]

14 Nov 2024

PONE-D-24-44009Burden of disease, incidence, and spatial distribution of bipolar disorder in Ecuador from 2011 to 2021 using hospital discharge recordsPLOS ONE

Dear Dr. MASCIALINO,

Thank you for submitting your manuscript to PLOS ONE. After careful consideration, we feel that it has merit but does not fully meet PLOS ONE’s publication criteria as it currently stands. Therefore, we invite you to submit a revised version of the manuscript that addresses the points raised during the review process.

We look forward to receiving your revised manuscript.

Kind regards,

Ramesh Athe, PhD

Academic Editor

PLOS ONE

2. We note that Figures 3 and 4 in your submission contain [map/satellite] images which may be copyrighted. All PLOS content is published under the Creative Commons Attribution License (CC BY 4.0), which means that the manuscript, images, and Supporting Information files will be freely available online, and any third party is permitted to access, download, copy, distribute, and use these materials in any way, even commercially, with proper attribution. For these reasons, we cannot publish previously copyrighted maps or satellite images created using proprietary data, such as Google software (Google Maps, Street View, and Earth). For more information, see our copyright guidelines: http://journals.plos.org/plosone/s/licenses-and-copyright.

1. You may seek permission from the original copyright holder of Figures 3 and 4 to publish the content specifically under the CC BY 4.0 license. 

Reviewers' comments:

Reviewer's Responses to Questions

**Comments to the Author**

1. Is the manuscript technically sound, and do the data support the conclusions?

Reviewer #1: Partly

Reviewer #2: Yes

2. Has the statistical analysis been performed appropriately and rigorously? 

Reviewer #1: No

Reviewer #2: Yes

3. Have the authors made all data underlying the findings in their manuscript fully available?

Reviewer #1: Yes

Reviewer #2: Yes

4. Is the manuscript presented in an intelligible fashion and written in standard English?

Reviewer #1: Yes

Reviewer #2: Yes

5. Review Comments to the Author

Reviewer #1: 1. Title and Introduction:

* Title: The title is clear and reflects the paper’s content, but it could be made more concise and engaging.

* Introduction: The introduction provides a good background on bipolar disorder and highlights the knowledge gap, but more emphasis on previous local studies would strengthen the rationale for the research.

* Suggestions:

- Add a short paragraph about the economic impact of bipolar disorder in Latin America to bolster the study’s justification.

- Incorporate more up-to-date references to support the background.

2. Methods:

* The methods are well-documented, but there is limited discussion about how differences between public and private hospital records were handled.

*No adjustment for other demographic factors like social class or education was mentioned.

* Suggestions:

- Include a deeper analysis of additional factors like geographic region or socioeconomic status to provide a more comprehensive view.

- Explain the steps taken to ensure the quality of hospital records and how repeated hospitalizations for the same patients were handled.

3. Results:

* Suggestions:

- A comparative analysis with other Latin American countries would highlight regional differences.

4. Discussion:

* The discussion touches on key findings, but it leans heavily on the statistics without sufficiently tying them to practical applications or health policy.

* There is little mention of gaps that future studies could address.

* Suggestions:

- Strengthen the discussion by proposing specific public health interventions or policy recommendations based on the findings.

- Elaborate more on the study’s limitations and how future research could overcome them, such as improving hospital record systems.

5. palgiarism:

*Plagiarism Check Result: 38.14% (By Plagiarism Detector v. 2695)

(It is recommended to reduce this percentage as much as possible to ensure originality).

Reviewer #2: Burden of disease, incidence, and spatial distribution of bipolar disorder in Ecuador from 2011 to 2021 using hospital discharge records

PONE-D-24-44009

Comments

Overall this research appears to be very well-conceived and conducted!

I believe that the paper will have value at both a local and global level. This contributes to our understanding of the specific situation in Ecuador in a large way and to the global burden of bipolar disorder as well.

Recommended for publication

Minor revision suggestions follow:

Is the manuscript technically sound, and do the data support the conclusions?

The manuscript is very well written and easy to follow.

In fact, it was enjoyable and informative to read.

The conclusions are robust and follow logically from the data and analysis, yet are not overstated.

Limitations of the research are identified and the conclusions fall within the scope of the research.

Has the statistical analysis been performed appropriately and rigorously

The statistical analysis has been conducted to a high quality in line with accepted practice.

Note Consider for revision:

Line 140:

A 3% discount rate was used. This is in line with the standard discounting practice.

As Ecuador is identified by the authors as a middle to low-income country, consider running the model with a second analysis using a 5% discount rate. This may affect the high contribution of YLDs to DALYs identified on line 243.

Table 2.

Headings: General, Public Sector, Private Sector.

Consider changing "General" to "Weighted Average". Is this correct?

Have the authors made all data underlying the findings in their manuscript fully available?

The data used are publicly available.

Is the manuscript presented in an intelligible fashion and written in standard English?

The manuscript is very well written in terms of both language and content!

The following minor suggestions are offered:

Line 182 change: "ones" to "cantons"

Line 191 change: "inferior" to "below"

Line 239 change: "attention" to "treatment" (This one is difficult)

Line 306 change: "the bipolars disorder burden" to "the burden from bipolar disorder"

Figure 4. Could also benefit from more labelling for those who are less familiar with Ecaudore's geography.

6. PLOS authors have the option to publish the peer review history of their article (what does this mean? ). If published, this will include your full peer review and any attached files.

**Do you want your identity to be public for this peer review?** For information about this choice, including consent withdrawal, please see our Privacy Policy .

Reviewer #1: **Yes: ** Sameh Tamer Moustafa Awad Eldaly

Reviewer #2: **Yes: ** Francesco Bolstad

---

## [Author Response · Author response to Decision Letter 1]

30 Jan 2025

Ramesh Athe, PhD

Academic Editor

PLOS ONE 

We have carefully reviewed and ensured that the manuscript fully complies with PLOS ONE's style requirements. All aspects, including file format, font, headings, page and line numbers, reference style, manuscript organization, and file naming, have been revised and adjusted according to the journal's guidelines:

A rebuttal letter that responds to each point raised by the academic editor and reviewer(s). This file is labeled as 'Response to Reviewers'.

A marked-up copy of the manuscript that highlights changes made to the original version. This file is labeled as 'Revised Manuscript with Track Changes'.

An unmarked version of the revised paper without tracked changes. This file is labeled as 'Manuscript'.

We have taken special care to ensure consistency and adherence throughout the submission.

All required files were submitted according to the guidelines.

2. We note that Figures 3 and 4 in your submission contain [map/satellite] images which may be copyrighted. All PLOS content is published under the Creative Commons Attribution License (CC BY 4.0), which means that the manuscript, images, and Supporting Information files will be freely available online, and any third party is permitted to access, download, copy, distribute, and use these materials in any way, even commercially, with proper attribution. For these reasons, we cannot publish previously copyrighted maps or satellite images created using proprietary data, such as Google software (Google Maps, Street View, and Earth). For more information, see our copyright guidelines: http://journals.plos.org/plosone/s/licenses-and-copyright.

We require you to either (1) present written permission from the copyright holder to publish these figures specifically under the CC BY 4.0 license, or (2) remove the figures from your submission.

Dear PLOS Staff, regarding figures 3 and 4, we have made the following adjustments in the document to acknowledge the origin of open data used in the creation of these:

The maps were designed and created by the authors. The shape files used in this study were collected from the INEC portal (https://www.ecuadorencifras.gob.ec/documentos/web-inec/Geografia_Estadistica/Micrositio_geoportal/index.html) according to their licensing requirements (https://www.ecuadorencifras.gob.ec/institucional/politica-datos-personales/). Continental and country borders were Made using shapefiles obtained from Natural Earth portal http://www.naturalearthdata.com/about/terms-of-use/ .

The licensing of these data complies with PLOS ONE policy:

“Any maps included or created as part of a figure must use a basemap tile, shapefile, or image compatible with our CC BY 4.0 license. The basemap refers to the foundational geographic layer of the map (possibly including country boundaries, for example) onto which other layers of data are plotted. Satellite and aerial images may also be used as basemaps.

If your submission file inventory includes a map, we will ask you to provide a direct link to the source of the basemap and provide attribution to this source in the corresponding figure legend. We will also ask you to provide information regarding the terms of use or license information for the map.

If you created the map in a software program like R or ArcGIS, please locate the source of the basemap within the package used to generate the map.”

3. Please include a separate caption for each figure in your manuscript

Thank you for pointing out the need for separate captions for each figure in the manuscript. We have carefully reviewed and updated the document to ensure that each figure now includes a clear and independent caption. All captions have been added to the corresponding part of the manuscript.

Fig 1. Time trend of bipolar disorder hospitalized yearly incidences male and female cases (2010-2021)

The graph shows the annual incidence of hospitalizations for bipolar disorder per 100,000 person-years, with separate trends for males (blue line), females (red line), and the total population (green line). Female incidence rates are consistently higher than those of males across the entire period. The total incidence follows a similar trend to females, reflecting their predominant contribution. A notable peak is observed in 2019, followed by a decline in subsequent years. These trends provide insights into sex-based differences in hospitalization rates for bipolar disorder over time.

Fig 2. Distribution of Bipolar disorder hospital admission by age (total, female and male)

The histograms show the number of hospitalized cases across age groups for the total population (bottom), females (top-left), and males (top-right). All distributions demonstrate a peak in hospitalization rates among individuals aged 20–40 years, with a gradual decline in older age groups. The female distribution shows a slightly higher number of cases compared to males, particularly in younger and middle-aged groups, reflecting sex-based differences in hospital admissions for bipolar disorder.

Fig 3. Spatial clusters of bipolar disorder in Ecuador.

This figure illustrates spatial clusters of bipolar disorder incidences across Ecuador, identifying regions with significantly higher or lower incidence rates. Clusters were determined using SaTScan, based on data from 2010-2021. Single cluster cantons with statistically significant clusters are marked in red dots for high-incidence clusters, grey circles describe high incidence larger areas including more than one single canton cluster. The shape files used in this figure for province and cantonal borders were obtained from the INEC portal ( https://www.ecuadorencifras.gob.ec/documentos/web- inec/Geografia_Estadistica/Micrositio_geoportal/index.html ). Continental and country borders were Made using shapefiles obtained from Natural Earth portal http://www.naturalearthdata.com/

Fig 4. Incidences of bipolar disorder in Ecuador (2010-2021).

The figure depicts the distribution of bipolar disorder incidences across different regions of Ecuador for the 2010-2021 period. The shape files used in this figure for province and cantonal borders were obtained from the INEC portal ( https://www.ecuadorencifras.gob.ec/documentos/web- inec/Geografia_Estadistica/Micrositio_geoportal/index.html ). Continental and country borders were Made using shapefiles obtained from Natural Earth portal http://www.naturalearthdata.com/

We believe this modification improves the clarity and overall presentation of the manuscript, as requested.

Reviewers' comments:

Reviewer #1:

1. Title and Introduction:

* Title: The title is clear and reflects the paper’s content, but it could be made more concise and engaging.

Thank you for your suggestion. The new title is: “Trends, Geographic Distribution, and Disease Burden of Bipolar Disorder in Ecuador (2011–2021): An Analysis of Hospital Discharge Data.”

* Introduction: The introduction provides a good background on bipolar disorder and highlights the knowledge gap, but more emphasis on previous local studies would strengthen the rationale for the research.

* Suggestions:

- Add a short paragraph about the economic impact of bipolar disorder in Latin America to bolster the study’s justification.

Thank you for your comment. We included a substantial paragraph regarding the economic impact of bipolar disorder in Latin America in the introduction.

- Incorporate more up-to-date references to support the background.

Thank you for your valuable suggestion. We have carefully reviewed and updated the references in the manuscript, replacing outdated sources with more recent and relevant studies (from reference 3 to 13). The revised references provide up-to-date support for the background.

2. Methods:

* The methods are well-documented, but there is limited discussion about how differences between public and private hospital records were handled.

Thank you for pointing this omission out. We have now provided a more complete description of the public and private institutions and how they have been determined through the data set. For ease of reference, the addition is: “The database provided details of the sources of hospital record information categorized by the type of institution. The public sources listed include Seguro Social (Social Security), the Instituto Ecuatoriano de Seguridad Social (Ecuadorian Social Security Institute), the Junta de Beneficencia de Guayaquil (Guayaquil Charitable Board), the Ministerio de Defensa Nacional (Ministry of National Defense), the Ministerio de Educación (Ministry of Education), the Ministerio de Gobierno y Policía (Ministry of Government and Police), the Ministerio de Justicia y de Gobierno y Policía (Ministry of Justice and Government and Police), the Ministerio de Justicia, Derechos Humanos y Cultos (Ministry of Justice, Human Rights, and Worship), the Ministerio de Salud Pública (Ministry of Public Health), municipalities, other public institutions, Seguro Campesino (Rural Social Security), and universities and polytechnic schools. The private sources are grouped into two categories: private institutions for profit and private institutions not for profit”.

*No adjustment for other demographic factors like social class or education was mentioned.

* Suggestions:

- Include a deeper analysis of additional factors like geographic region or socioeconomic status to provide a more comprehensive view.

We agree that a more profound analysis of the intersection of socio-economic factors with the cost and distribution of bipolar disorder would be a very useful contribution to the literature, and more particularly to those involved in policy-making. Unfortunately, the hospital record data available do not contain information on relevant socio-economic variables such as income, education, family structure etc. It does link hospital records to geographic regions, however, and we have included a geographic analysis of the epidemiological burden of disease of bipolar disorder in the paper. Despite identifying statistically significant spatial clusters, no clear pattern attributable to socioeconomic factors was observed. The clusters were located in distinct areas of the northern and southern highlands of Ecuador, which have highly heterogeneous socioeconomic characteristics, as well as in a coastal cluster within Ecuador's main port city. Therefore, any analysis on socioeconomic factors with the current data would be speculative and lack objectivity. However, identifying spatial clusters of bipolar disorder within the country is highly valuable, as they can serve as indicators for further research in the identified areas and highlight the need for geographically targeted interventions.

- Explain the steps taken to ensure the quality of hospital records and how repeated hospitalizations for the same patients were handled.

Additional information regarding policies that ensure data quality was added to the methods section. Unfortunately, as the hospital data records are anonymous, there is no way to control for repeated hospitalization. We noted this limitation in the methods section as well.

3. Results:

* Suggestions:

- A comparative analysis with other Latin American countries would highlight regional differences.

Thank you for your suggestion. Given that the dataset did not have information about other Latin American countries, we could not conduct a comparison in the results section. However, a comparison with other countries in the region was added to the discussion section.

4. Discussion:

* The discussion touches on key findings, but it leans heavily on the statistics without sufficiently tying them to practical applications or health policy.

We appreciate your valuable feedback. In response to your suggestion, we have revised and adjusted the discussion section to strengthen the connection between the presented statistics and their potential practical applications, as well as their relevance to health policy. In this regard, we have included concrete examples that contextualize the data within practical scenarios and the design of public health policies. We hope these modifications enhance the integration between the statistical conclusions and their practical applicability.

- Strengthen the discussion by proposing specific public health interventions or policy recommendations based on the findings.

We appreciate your observation. To address your suggestion, we have expanded the discussion section by incorporating specific recommendations for public health interventions and policies based on the study's findings. We believe these additions enrich the content and strengthen the connection between the findings and their practical application.

- Elaborate more on the study’s limitations and how future research could overcome them, such as improving hospital record systems.

Thank you for your valuable suggestion. The limitations section was expanded and enriched as per your suggestions.

5. plagiarism:

*Plagiarism Check Result: 38.14% (By Plagiarism Detector v. 2695)

(It is recommended to reduce this percentage as much as possible to ensure originality).

Thank you for your valuable feedback regarding the similarity percentage identified by Plagiarism Detector. Following your suggestion, we have carefully reviewed the manuscript and addressed the issue. A significant portion of the similarity arises from a preprint under review of another article on a related topic (Mascialino, G., Carrington, S. J., Coral-Almeida, M., Álvarez-Mejía, D., Torres-Tapia, M. E., & Rodríguez-Lorenzana, A. (2023). Burden of disease, incidence, and spatial distribution of Schizophrenia in Ecuador from 2011 to 2021 using hospital discharge records), specifically in the method section, which does not vary in this type of study and the program flagged as overlapping content. Additionally, affiliations, names of public institutions (e.g., “Instituto de Seguridad Social”), technical terms (e.g., “Years Lived with Disability” or “Global Burden of Disease”), references and similar elements account for approximately 10% of the similarity index, which cannot be changed.

The remaining content has been thoroughly revised and adjusted to minimize overlap and ensure originality. We appreciate your attention to this matter and remain committed to maintaining the highest standards of academic integrity.

Reviewer #2: Burden of disease, incidence, and spatial distribution of bipolar disorder in Ecuador from 2011 to 2021 using hospital discharge records

Note Consider for revision:

Line 140:

A 3% discount rate was used. This is in line with the standard discounting practice.

As Ecuador is identified by the authors as a middle to low-income country, consider running the model with a second analysis using a 5% discount rate. This may affect the high contribution of YLDs to DALYs identified on line 243.

Thank you for your valuable insights and comments regarding the appropriate discount rate. This is indeed a crucial topic, as the choice of discount rate determines the intertemporal value assigned to health outcomes over time. Ultimately, it shapes the prioritization of health investments, especially in resource-scarce settings, and serves as a key tool for evaluating whether a particular intervention represents the best allocation of societal resources. For instance, comparing two distinct streams of costs and benefits over time requires the application of a well-justified discount rate to determine which option yields better returns for society.

We wholeheartedly agree with the importance of applying the correct discount rate to facilitate comparisons within a specific context, such as within a country where decisions between competing options are made. An overly high discount rate could lead to the rejection of socially desirable projects, while an exce

---

## [Editor Report · Decision Letter 1]

18 Feb 2025

Trends, Geographic Distribution, and Disease Burden of Bipolar Disorder in Ecuador (2011–2021): An Analysis of Hospital Discharge Data

PONE-D-24-44009R1

Dear GUIDO MASCIALINO,

We’re pleased to inform you that your manuscript has been judged scientifically suitable for publication and will be formally accepted for publication once it meets all outstanding technical requirements.

Kind regards,

Ramesh Athe, PhD

Academic Editor

PLOS ONE
---

## [Editor Report · Acceptance letter]

PONE-D-24-44009R1

PLOS ONE

Dear Dr. MASCIALINO,

I'm pleased to inform you that your manuscript has been deemed suitable for publication in PLOS ONE. Congratulations! Your manuscript is now being handed over to our production team.

Kind regards,

on behalf of

Dr. Ramesh Athe

Academic Editor

PLOS ONE